# Lateral Root Initiation and the Analysis of Gene Function Using Genome Editing with CRISPR in *Arabidopsis*

**DOI:** 10.3390/genes12060884

**Published:** 2021-06-08

**Authors:** Nick Vangheluwe, Tom Beeckman

**Affiliations:** 1Department of Plant Biotechnology and Bioinformatics, Ghent University, 9052 Ghent, Belgium; Nick.Vangheluwe@psb.ugent.be; 2VIB Center for Plant Systems Biology, 9052 Ghent, Belgium

**Keywords:** lateral root initiation, asymmetric cell division, auxin signalling, CRISPR, TSKO, genome editing, *Arabidopsis*, gene function

## Abstract

Lateral root initiation is a post-embryonic process that requires the specification of a subset of pericycle cells adjacent to the xylem pole in the primary root into lateral root founder cells. The first visible event of lateral root initiation in *Arabidopsis* is the simultaneous migration of nuclei in neighbouring founder cells. Coinciding cell cycle activation is essential for founder cells in the pericycle to undergo formative divisions, resulting in the development of a lateral root primordium (LRP). The plant signalling molecule, auxin, is a major regulator of lateral root development; the understanding of the molecular mechanisms controlling lateral root initiation has progressed tremendously by the use of the *Arabidopsis* model and a continual improvement of molecular methodologies. Here, we provide an overview of the visible events, cell cycle regulators, and auxin signalling cascades related to the initiation of a new LRP. Furthermore, we highlight the potential of genome editing technology to analyse gene function in lateral root initiation, which provides an excellent model to answer fundamental developmental questions such as coordinated cell division, growth axis establishment as well as the specification of cell fate and cell polarity.

## 1. Introduction

### 1.1. Arabidopsis thaliana as a Model Plant to Study Root Development

The *Arabidopsis* plant, as with most angiosperms, develops an extensive root system designed to function in the anchorage of the plant, in the absorption of water and mineral ions and in interaction with microorganisms. Several properties make roots amenable to developmental studies: the root apical meristem is accessible and not embedded in developing organs or primordia; the root contains no pigment and is therefore essentially transparent; and there are relatively few differentiated cell types in roots. In addition, root morphogenesis in many plants occurs in a continuous and relatively uniform pattern without significant developmental transitions, while cell files are easy to observe in longitudinal sections and their origin can be traced back to the meristem [1]. Understanding of root morphology and development in *Arabidopsis* has largely originated from studies of the seedling root system.

The remarkably simple anatomy of the *Arabidopsis* primary root has its origin in the embryonic root. Upon germination, the cells in the root meristem initiate a program of regulated cell division and expansion. Since there are no morphogenetic cell movements in plants, the final form of the root is primarily controlled by three parameters: the timing of cell division, the orientation of the plane of cell division and the degree and direction of cell expansion. The ability of a root to grow in a continuous fashion is dependent on the regulation of cell division and expansion as well as maintenance of a stem cell population within the meristem. The ultimate architecture of the plant root system depends on environmental conditions as well as genetic factors. Root growth can be profoundly affected by a variety of external stimuli, including gravity, light, temperature, moisture, aeration and physical obstacles [2]. These stimuli can alter cell division activity, the direction or degree of cell expansion, the amount of root branching, or the structure of root cells.

### 1.2. Lateral Root Development Enables Root Branching

Root branching is commonly known to occur by the formation of lateral roots, roots formed from internal layers along the parent root axis. However, the first rooting plant lineage, lycophytes, was not able to generate lateral roots. Instead, dichotomous branching of the root tip, involving the formation of two new root apical meristems from two apical stem cells, allowed these plants to shape their root system architecture [3]. Hence, lycophytes are able to have root branches only at the root tips. Evolution of root branching in later diverging plant lineages is accompanied by an increase in plasticity [4]. Some ferns have a fixed number of lateral root stem cells specified within clonally related groups of cells derived from the daughter cells of the apical cell and referred to as merophytes that are maintained along the root and are competent to form lateral roots [5,6,7,8]. In the model fern *Ceratopteris richardii*, two out of three successive merophytes have the competence to form a lateral root, resulting in a regular branching pattern [5,9]. Hence, lateral roots of ferns contribute to an increased capacity to explore the substrate as compared to terminal branching roots in lycophytes, but their fixed positioning still restricts the plasticity of their root system.

### 1.3. In Arabidopsis Lateral Roots Arise from the Pericycle

In seed plants, lateral roots are initiated endogenously along the main root axis from a specific subset of pericycle cells, also called lateral root founder cells [10,11,12]. The pericycle is composed of two different types of cells—cells located in front of the two phloem poles and cells situated in front of the two xylem poles—each with different cytological features and cell fates [13,14,15,16]. Remarkably, *Arabidopsis* xylem pole pericycle cells are in a division-competent stage while being part of the differentiated part of the primary root [13,17]. In addition, these cells display physiological and genetic characteristics that resemble those of root meristem cells and can be the source of massive induction of lateral roots [16,18]. Understanding of the mechanisms controlling lateral root development has progressed tremendously through studies in *Arabidopsis*, which were recently reviewed in Banda et al. (2019) and Du and Scheres (2018). Lateral root formation can be divided into four steps: lateral root positioning, lateral root initiation, lateral root development and patterning and lateral root emergence [19]. Newly formed lateral roots consist of de novo patterned root tissues and meristems, resembling those in the primary roots that ensure their indeterminate growth.

### 1.4. Initiation of Lateral Roots Is Marked by Coordinated Migration of Nuclei and Cell Divisions

The first visible event of lateral root initiation is the simultaneous migration of nuclei of neighbouring pericycle lateral root founder cells towards the common cell wall, followed by an asymmetric anticlinal cell division, giving rise to two small daughter cells and two larger flanking cells, which is referred to as a Stage I LRP [11,12] (Figure 1). Asymmetric cell divisions are formative divisions that generate daughter cells of distinct identity and are essential in enabling post-embryonic organogenesis [20]. In mutants with impaired lateral root formation, no simultaneous polar movement of nuclei in lateral root founder cells could be observed [11]. These observations revealed that the coordinated nuclear migration of two neighbouring xylem pole pericycle nuclei might be a prerequisite for proper primordium initiation and the formation of lateral roots.

The next division occurs periclinally and yields a two-layered (Stage II) LRP [12]. Subsequently, a series of anticlinal and periclinal cell divisions and differentiation steps leads to cell diversity and tissue patterns, resulting in the development of a dome-shaped LRP that progressively acquires the same tissue organisation as the root meristem and eventually emerges through overlying tissues of the primary root. Distinct LRP stages (I–VII) have been classified based on anatomical analysis, considering the number of cell-layers that the LRP comprises or based on its position through the overlaying tissues [12,21]. Intriguingly, it was shown that the tissues in the primary root overlaying an LRP influence the shape and development of the primordium [22,23,24]. The growing LRP needs to deal with the mechanical constraints imposed by the surrounding tissues. Endodermal cells need to change shape and lose volume to accommodate the expansion of the LRP. It was shown that endodermal feedback is already required very early for the execution of the first formative divisions and later, for the growth of the LRP through this persistent cell layer [23].

### 1.5. Lateral Root Founder Cells Are Specified in the Pericycle

*GATA23* is the earliest known marker for lateral root founder cell specification that was identified by meta-analysis of transcriptomic datasets for lateral root initiation [11,25]. *GATA23* is expressed in xylem pole pericycle cells before the first asymmetric division. Moreover, it was shown that *GATA23* expression is controlled by an auxin signalling mechanism [11]. Xylem pole pericycle cells pass through a developmental window for lateral root initiation in which, at minimum auxin concentration, these cells have a high probability of becoming specified founder cells [10]. The endodermis assists in the transition from the founder cell stage to the lateral root initiation phase via an auxin reflux pathway between endodermal cells and the adjacent founder cells [26,27]. Next, when a local auxin concentration maximum is reached, several auxin signalling components interact together and the founder cells proceed to lateral root initiation [11,28].

## 2. Auxin Signalling Is Essential for Lateral Root Development

Auxin acts as a common integrator to many endogenous and environmental signals, regulating lateral root development [29]. A plethora of auxin ‘signalling modules’ act sequentially during lateral root development and control various steps of lateral root formation, from priming to initiation, patterning, and emergence (Figure 2). An auxin response module is defined as a pair of strongly interacting Aux/IAA proteins and Auxin Response Factors (ARFs), which, together, regulate a subset of primary auxin response genes [30]. The properties of this auxin response depend on the cellular auxin concentration, F-box (TIR1 and AFB1-5) affinity for auxin and for the Aux/IAA target protein, Aux/IAA–ARF interaction, as well as ARF activity and affinity for the promoter of its target genes.

Developmental decisions on the distribution of lateral roots already take place in the distal zone of the primary root tip, in a transition zone between the apical meristem and the elongation zone, also formerly referred to as the basal meristem. Studies using *DR5*-based reporters suggest that periodic auxin response, along with oscillating waves of gene expression, functions as an endogenous clock-like mechanism [11,28,31,32,33,34,35]. Following an auxin response maximum in the protoxylem cell file, the neighbouring xylem pole pericycle cells are ‘primed’ and form pre-branch sites, providing them with the competence to develop lateral roots. The mechanism through which the ‘priming information’ is transmitted from the protoxylem to the overlaying xylem pole pericycle has not been characterised yet; however, computational modelling combined with in planta experiments recently led to the insight that both the frequency of priming and the passing on of the auxin maximum from protoxylem to pericycle cells can be explained by growth dynamics in the meristem periodically, generating cell size and hence auxin loading potential variations [42,43]. Concerning the auxin response mechanism involved, it was shown that expression of *GATA23* is dependent on the IAA28–ARF7/19 auxin module [11]. In addition, transactivation of *GATA23* in xylem pole pericycle cells in the *iaa28* gain-of-function mutant is able to rescue the dominant *iaa28* lateral root mutant phenotype [11]. These observations suggest that an IAA28-dependent auxin signalling mechanism controls *GATA23* expression, regulating lateral root founder cell specification prior to lateral root initiation. Furthermore, it was shown that ARF7 together with its auxin-sensitive inhibitor IAA18/POTENT would be responsible for inducing a negative regulatory loop contributing to the oscillatory pattern in the auxin response [32].

Auxin regulates and coordinates both founder cell divisions and founder cell polarity/identity specification during lateral root initiation. The auxin response increases in founder cell pairs a few hours before initiation of lateral root organogenesis occurs [10]. Disturbing the auxin response in founder cells using polar auxin transport inhibition is sufficient to block initiation, whereas artificial auxin production in single xylem pole pericycle cells triggers initiation, indicating that auxin accumulation is necessary and sufficient for lateral root initiation [10,44]. Moreover, coordinated regulation of auxin influx and efflux carriers including AUXIN RESISTANT 1 (AUX1) and PIN-FORMED (PIN) in the LRP and surrounding tissues is needed to establish the auxin gradient essential for lateral root initiation (Figure 2) [26,27,45].

In *solitary-root* (*slr*)*-1*, a dominant negative mutant of *SLR*/*IAA14*, lateral root initiation is blocked at the G1-to-S transition and no nuclear migration in paired founder cells can be observed. Moreover, the defects in *slr-1* cannot be restored by the application of exogenous auxin. The *arf7arf19* double mutant phenocopies *slr-1* and IAA14 interact with ARF7 and ARF19, indicating that auxin stimulates lateral root initiation through the SLR/IAA14–ARF7,19 signalling module (Figure 3). Following the first asymmetric division, the small daughter cells exhibit an auxin maximum, which is accompanied by BODENLOS/IAA12–MONOPTEROS/ARF5-dependent signalling [46]. It was shown that the hemizygous gain-of-function *bdl* mutants and weak loss-of-function *monopteros* (*mpS319*) mutants display abnormalities in pericycle divisions and lateral root positioning [46]. Taken together, a second auxin signalling module involving BDL/IAA12 and MP/ARF5 regulates lateral root initiation together with SLR–ARF7–ARF19 (Figure 3). It has been proposed that auxin plays an instructive role in the structural and functional patterning of the LRP similar to the shoot and root apical meristems [26]. However, exactly how this auxin gradient actually governs cell identities and divisions is still poorly understood.

## 3. The Cell Cycle Drives Lateral Root Initiation

The pericycle in *Arabidopsis* is a heterogeneous tissue with diarch symmetry composed of two cell types with distinct cell division ability [16]. Phloem pole pericycle cells are mitotically inactive, whereas xylem pole pericycle cells retain stem cell activity after leaving the primary root meristem and thus, maintain a division-competent state essential for lateral root formation [13,14,16]. It was shown that the nuclear protein ABERRANT LATERAL ROOT FORMATION 4 (ALF4) is required to maintain xylem pole pericycle cells in a mitosis-competent state [10,48,49]. The *alf4-1* mutant has a normal primary root, but is deficient in lateral root initiation. Although auxin is required at this stage of the process, it was proposed that ALF4 functions independently of auxin (DiDonato et al., 2004). Recent discoveries reveal that *alf4* is resistant to auxin and displays reduced *DR5* activity, suggesting that reduced auxin response contributes to the lateral root defect [50]. Furthermore, it was shown that ALF4 physically interacts with RING BOX (RBX1), a subunit of the SCF^TIR1^ complex regulating the degradation of Aux/IAA proteins [50]. The *alf4* mutant stabilizes the SCF^TIR1^ substrate IAA17, and SCF^TIR1^-dependent degradation of Aux/IAA proteins is inhibited by ALF4, which suggests that increased levels of SLR/IAA14 in the *alf4* mutant could result in reduced auxin response during lateral root formation [50].

The onset of lateral root initiation coincides with the occurrence of a series of anticlinal, asymmetric divisions in the xylem pole pericycle. Hence, cell cycle activation is inherently connected with lateral root initiation. The transition of xylem pole pericycle cells from G1 to S and subsequent cycle progression are stimulated by auxin. These ‘primed’ cells reactivate the cell cycle only when they reach the lateral root initiation zone, which indicates that activating cell cycle-related genes alone is not sufficient to initiate a new lateral root [39,51]. Moreover, disturbing the auxin response through inhibition of polar auxin transport or impaired auxin signalling is sufficient to inhibit the cell divisions necessary for lateral root initiation [18,39,44]. Activation and progression through the major phases of the cell cycle are governed by the control of Cyclin-dependent kinases (CDKs). Several highly conserved components of the cell cycle have been demonstrated to be important for lateral root initiation [20]. For instance, Lateral Organ Boundaries Domain (LBD) 18 and LBD33 lateral root regulatory protein dimers mediate lateral root initiation by direct binding to the promoter of *E2Fa*, which encodes a transcriptional activator of cell cycle genes [52]. *E2Fa* is expressed during lateral root initiation and promotes the first asymmetric cell divisions [46,52].

Recently, it has been demonstrated that the lateral root regulatory *PICKLE* gene encoding a chromatin remodelling factor interacts with *Retinoblastoma-related 1* (RBR1) to repress *LBD16* promoter activity [53]. *RBR1* Is expressed in xylem pole pericycle cells and partially silencing *RBR1* expression results in increased root branching. Inhibition of LR formation by PKL–RBR1 is counteracted by auxin, indicating that, in addition to auxin-mediated transcriptional responses, the fine-tuned process of LR formation is also controlled at the chromatin level in an auxin-signalling dependent manner [53].

The auxin-mediated G1 to S transition is inhibited by the Interactor of CDK/Kinase inhibitory protein (KIP)-related protein (ICK/KRP) family of proteins, hence preventing lateral root initiation [18,54]. Loss-of-function mutants of *KRP2* display increased lateral root density, whilst overexpression of *KRP2* results in a large reduction in lateral root density [18,54,55]. KRP2 interacts with the CDKA;1–CYCD2;1 complex and results in accumulation in the nucleus of the inactive complex [54]. Upon auxin treatment, reduced *KRP2* expression and increased KRP2 protein turnover result in a transient increase in CDKA;1–CYCD2;1 activity and subsequent cell division, which promotes lateral root initiation [55]. Other D-type cyclins such as *CYCD4;1* and *CYCD3;1* are also shown to be involved in lateral root initiation [18,56]. In addition, A2-type cyclins are involved in early G2 to M transition of the cell cycle during lateral root initiation. The triple *cyca2;234* mutant displays a delay in the expression of mitotic regulators, while auxin signalling and G1 to S regulatory genes remain unaffected [57].

The F-box protein S-PHASE KINASE-ASSOCIATED PROTEIN 2A (SKP2A) positively regulates lateral root initiation [58,59]. Auxin binds directly to SKP2A and mediates the proteolysis of cell cycle-repressing transcription factors in a TIR1-AFB auxin receptor-independent pathway. Overexpression of *SKP2A* in the *tir1* mutant induces lateral root initiation and *skp2a* mutants display an auxin-resistant root growth phenotype. In contrast, a close homologue, SKP2B, negatively regulates the cell cycle and lateral root development as it represses founder cell divisions [60].

In summary, strict control of cell division is regulated by highly conserved inhibiting and activating components of the cell cycle and is required for lateral root organogenesis.

## 4. Genome Editing for Functional Genomic Studies in Lateral Root Development

### 4.1. Loss-of-Function Mutant Alleles Are Indispensable in Functional Genomic Studies

In view of the complexity of molecular control on lateral root initiation, the high number of potential regulators involved and the contribution of different tissue layers, solid genetic tools are a necessity to further unravel this process. Loss-of-function mutant alleles have been indispensable to analyse and demonstrate the function of genes in lateral root development. In plants, knockout or knockdown lines have been generated using various techniques such as ionizing radiation, ethyl methane sulfonate treatment, T-DNA or transposon insertions in the genome, RNA interference or artificial microRNAs. In addition, engineered nucleases can be used to generate knockout lines as a result of error-prone non-homologous end-joining (NHEJ) induced upon site-specific double-strand breaks in plant genomes. In the past five years, the generation of knockout plant lines via clustered regularly interspaced short palindromic repeats (CRISPR) genome editing technology has been widely adopted by researchers, while the basic principles behind double-strand break-induced targeted mutagenesis have been well known for decades [61]. Previous experiments demonstrated that by induction of double-strand breaks in genomes using a highly specific endonuclease, different types of genome editing can be achieved [61]. Distinct types of nucleases have been engineered including mega nucleases, zinc finger nucleases, transcription activator-like effector nucleases and CRISPR-associated (Cas) nucleases [62,63,64,65].

### 4.2. On the Origin of CRISPR

The CRISPR system originates from bacteria and archaea, in which it serves as an adaptive immune response system that degrades invading foreign plasmid or viral DNA [66]. The elucidation of the molecular mechanism of a type II CRISPR/Cas9 system from *Streptococcus pyogenes* has revealed a simple three-component system [64]. Cas9 is a nuclease that is able to cleave double-stranded DNA with two nuclease domains, each cleaving one of the two DNA strands. Target specificity is mediated by a short CRISPR RNA that binds directly to a stretch of 20 nucleotides on the target DNA, referred to as protospacer. An additional 3-nucleotide element termed protospacer-adjacent motif (PAM), with the sequence 5′-NGG-3′ downstream of the target sequence, is necessary for binding and cleavage by Cas9. This means that any 23-nucleotide sequence ending in 5′-GG-3′ can be targeted. The trans-activating CRISPR RNA interacts with the CRISPR RNA and facilitates the recruitment of Cas9, which results in the cleavage of the DNA target sequence 3 base pairs upstream of the PAM. Furthermore, it was shown that a direct fusion of the two RNAs to generate a chimeric guideRNA (gRNA) is functional as well [64].

### 4.3. Loss-of-Function Mutant Alleles in Arabidopsis Can Be Efficiently Generated with CRISPR

The first scientific report that described an effective CRISPR system to generate inheritable mutations in *Arabidopsis thaliana* was published in 2014 [67]. They used the constitutive *UBIQUITIN 4-2* promoter from *Petroselinum crispum* to drive Cas9 expression [67] and provided a Gateway^®^-based cloning system to clone up to two gRNA expression cassettes in the expression vector. Cas9 is very efficient in plants at inducing double-strand DNA breaks. Repair of DNA breaks by the error-prone NHEJ pathway ultimately results in the formation of short insertions and/or deletions (indels) at the break site [68]. These indels most often lead to frame shifts and/or early stop codons, which result in knockout mutations in the targeted gene(s). Currently, the most commonly used CRISPR system in plants is a two-component system based on Cas9 and the gRNA. However, many variations and applications have been developed which were recently reviewed in Wada et al. [69].

Most CRISPR efforts in plants to date have focused on generating stable and heritable mutant alleles for reverse genetics approaches, which has substantially contributed to the study of redundant gene families or genes for which no or a limited number of mutant alleles are available in *Arabidopsis* mutant collections [70,71,72,73]. However, this strategy is limited in case loss-of-function conveys severe pleiotropic phenotypes or even lethality. It is estimated that 10% of the approximately 25,000 protein-coding genes in the genome of *Arabidopsis* are essential [74]. Hence, detailed functional analysis of many fundamentally important plant genes is impeded and hinders the study of their function in a developmental-specific context.

### 4.4. Current Genetic Tools Comprise Certain Limitations for Functional Gene Studies in a Developmental-Specific Context

Lateral root development is a post-embryonic process that requires the specification of a subset of pericycle cells adjacent to the xylem pole in the primary root into lateral root founder cells [11]. Subsequently, during the process of lateral root initiation, cell fate specification and de novo lateral root meristem establishment are required for lateral root organogenesis [75,76,77]. These processes rely on key genetic players including *PIN*, *PLETHORA*, *AUX/IAA* and *ARF* genes that are necessary in primary root development as well [26,46,78,79,80,81]. Hence, a lot of loss-of-function alleles affect primary and lateral root development, which hampers functional analysis. Moreover, primary roots are initiated during embryogenesis and post-embryonic functional analysis of some of these common genetic players such as, for instance, *MP/ARF5* are limited because they govern essential functions during embryogenesis [78].

Different strategies have been pursued to enable a comprehensive investigation of gene function in specific developmental or physiological processes. An approach is the use of tissue-specific gene silencing [82,83]. However, gene silencing is often incomplete, interfering with the interpretation of the observed phenotypes, and it has been demonstrated that small RNAs can be mobile, limiting the tissue specificity in knockdown experiments [84]. Alternatively, transgenic vectors generating dominant-negative protein versions have been developed for certain genes and expressing these mutant versions in a tissue-specific context can locally interfere with endogenous gene functions [37,85]. Other methods include the conditional knockout of genes in specific cell types or tissues using Cre/lox-based clonal deletion [86,87]. However, these approaches rely on complicated genetic engineering and are difficult to scale.

### 4.5. CRISPR-TSKO Enables Lateral Root-Specific Loss-of-Function Studies

These limitations have been overcome using genome editing with CRISPR technology to generate conditional knockouts. Originally, tissue-specific promoters driving Cas9 expression have been employed, with the focus on increasing the chance of obtaining heritable mutant alleles [88,89,90]. For instance, the *NST3*/*SND1* promoter was used to drive xylem-specific Cas9 expression and target the essential gene *HCT* encoding a hydroxycinnamoyl transferase in *Arabidopsis* [91]. The potential of conditional gene knockouts of several essential genes in diverse plant cell types, tissues, and organs in *Arabidopsis* has recently been demonstrated [92]. Therefore, a versatile CRISPR tissue-specific knockout (CRISPR-TSKO) vector system was devised that allows for the specific generation of somatic DNA mutations in particular plant cell types, tissues, and organs [92,93,94]. Furthermore, an additional layer of conditionality was tested by integrating the CRISPR technology with an XVE-based, cell-type-specific inducible system [95,96,97]. This inducible CRISPR system in *Arabidopsis* enables efficient generation of target gene knockouts in desired cell types and at desired times [95].

Highly relevant for the molecular dissection of the lateral root formation program, it was recently shown that it is possible to specifically knockout genes in entire lateral roots using the promoter sequence of *GATA23* (Figure 3) [92]. As a proof-of-concept, *GFP* was targeted in *Arabidopsis* seedlings ubiquitously expressing *NLS-GFP* in the transgenic line *pHTR5:NLS-GFP-GUS* [92,98]. Fluorescence and sequence analysis of T1 and T2 seedlings demonstrated that organ-specific *GFP* knockout in lateral roots is highly efficient via the xylem-pole pericycle-expressed Cas9 controlled by *GATA23* (Figure 3). Interestingly, the observation that entire lateral roots lack a GFP signal provides evidence that *GATA23*-expressing precursor cells are clonally linked to the cells that constitute lateral roots.

In a next step, *ARF7* and *ARF19* were targeted as lateral root initiation is strongly inhibited in *arf7arf19* double mutants [92,99]. Surprisingly, lateral root initiation was only mildly affected when *ARF7* and *ARF19* were knocked out in *GATA23*-expressing pericycle cells [92], while an *arf7arf19* mutant is not capable of producing lateral roots. This suggests that the function of ARF7 and ARF19 in lateral root founder cells is not overpowering for lateral root development and raises the question of when and in which cells of the primary root these ARFs are crucial for lateral root organogenesis.

Recently, a CRISPR-based lateral root-specific repressor system was developed to study the function of genes by conditional knockdown [100]. In this system, a mutated version of Cas9 (dCas9) is used, which is no longer able to cleave DNA because of the perturbation of its nuclease activity. By targeting the dCas9 protein to the promoter regions of candidate genes, the resulting steric hindrance impacts the expression of the targeted genes. Interestingly, lateral root-specific simultaneous knockdown of *ARF7* and *ARF19* results in a mild reduction in lateral root density compared to the empty vector control [100]. This observation is consistent with lateral root-specific CRISPR-TSKO of *ARF7* and *ARF19*.

Functional analysis of *CDKA;1* and potential redundancy with other *CDK* members at the onset of lateral root initiation has been limited because loss-of-function of *CDKA;1* severely affects development [101]. Interestingly, lateral root branching is not affected when *CDKA;1* is knocked out in *GATA23*-expressing pericycle cells [92]. However, simultaneous knockout of *CDKA;1*, *CDKB1;1* and *CDKB1;2* halts lateral root growth soon after emergence with only a small number of lateral roots that arrest before emergence. These severely stunted lateral roots consist of a reduced number of extremely enlarged cells as a result of inadequate cell divisions and is reminiscent of mutants affected in cell cycle progression [101]. Taken together, *A*- and *B1*-subtype *CDK*s are concomitantly essential for lateral root organogenesis and lateral root-specific CRISPR-TSKO revealed that morphogenesis of lateral roots still occurs upon absence of intact cell cycle progression.

These experiments demonstrate that conditional knockouts enable the function of genes in spatial and temporal contexts of plant development to be studied and can pinpoint to unexpected and unexplored functions of known regulators. In summary, loss-of-function studies by generating inheritable or somatic mutations using genome editing opens avenues for discovering and analysing gene functions in lateral root development.

## 5. Conclusions

Root branching through lateral root formation is an important component of the adaptability of the root system to its environment. Regular spacing of lateral roots, as well as the initiation and development of lateral root primordia, is tightly regulated in *Arabidopsis*. However, lateral root development is readily influenced by external cues, ensuring the root system architecture is highly adaptable to different environmental conditions. To achieve such strict regulation while maintaining a high degree of flexibility, lateral root development relies on strong intercellular communication networks, mediated by the exchange of molecular messengers over both short and long distances.

Auxin acts as a common integrator to many endogenous and environmental signals regulating lateral root development. It was shown that auxin regulates and coordinates both lateral root founder cell divisions and founder cell polarity/identity specification during lateral root initiation. Thereafter, auxin plays an instructive role for the structural and functional patterning of the LRP. How and which molecular mechanisms auxin regulates during lateral root development are still poorly understood.

Root development is primarily controlled by three intertwined parameters: the timing of cell division, the orientation of the plane of cell division and the degree and direction of cell expansion. Hence, detailed functional analysis of genes involved in these fundamentally important processes in a developmental-specific context is limited because loss-of-function results in pleiotropic phenotypes or even embryo lethality. Lateral root-specific genome editing enables the analysis of gene function specifically in root organogenesis. The advantageous properties of *Arabidopsis* root development including simple morphology, small size, transparent organ combined with genome editing will undoubtedly contribute to a better understanding of these fundamental cellular processes.

## Figures and Tables

**Figure 1 genes-12-00884-f001:**
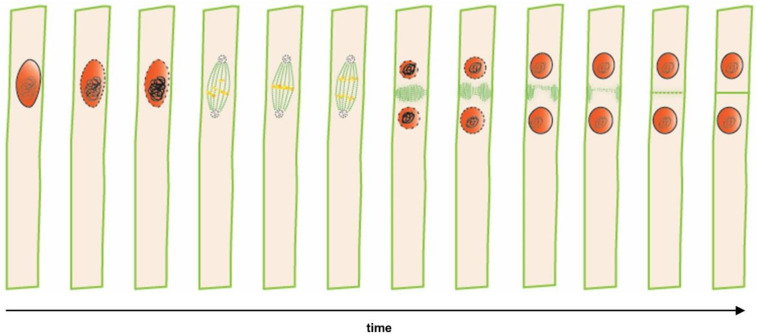
Time lapse from lateral root founder cell specification to asymmetric cell division in a single xylem pole pericycle cell. Sequential cellular events preceding the first asymmetric division that indicates nuclei (in **red**) and microtubules (in **green**). Polar migration of the nucleus in a lateral root founder cell is followed by an asymmetric anticlinal cell division, resulting in a short and a long daughter cell.

**Figure 2 genes-12-00884-f002:**
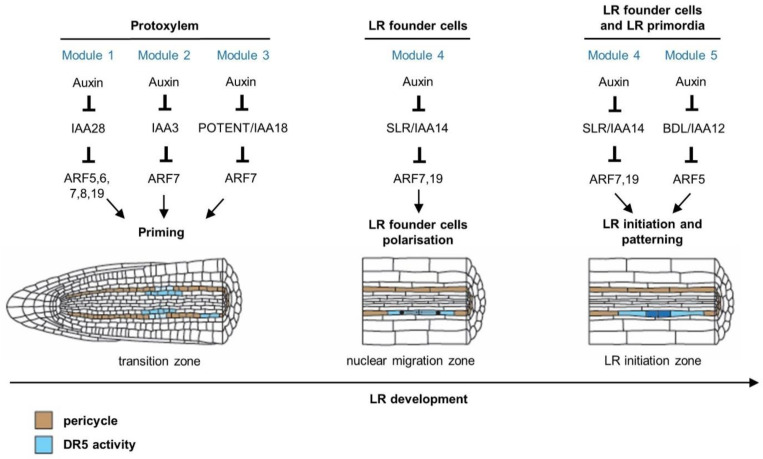
Auxin regulates different stages of lateral root development through multiple auxin-signalling modules in *Arabidopsis*. Lateral root founder cell priming involves the IAA28–ARF5, 6, 7, 8, 19, IAA3-ARF7 and POTENT/IAA18-ARF7 module (Modules 1, 2 and 3, respectively) in the basal meristem [11,31,32]. After priming, cells at prebranch sites maintain an increased auxin response, which was revealed through analysis of the *pDR5:LUCIFERASE* reporter [28,33,34,35]. Lateral root founder cells start to accumulate auxin, which triggers their polarisation and subsequent lateral root initiation [11]. The IAA14/SOLITARY-ROOT–ARF7-ARF19 module (Module 4) regulates the polarisation of lateral root founder cell pairs, which leads to coordinated nuclear migration towards the common cell walls [11,36]. Both the IAA14/SLR–ARF7,19 and the IAA12/BDL–ARF5 modules (Modules 4 and 5, respectively) are necessary for triggering lateral root initiation, which starts with an asymmetric anticlinal division of lateral root founder cells [37,38,39]. These modules also regulate the morphological and histological patterning of the LRP [26,40,41]. Cells coloured in blue indicate auxin response according to the synthetic *DR5* reporter. Cells coloured in brown belong to the pericycle.

**Figure 3 genes-12-00884-f003:**
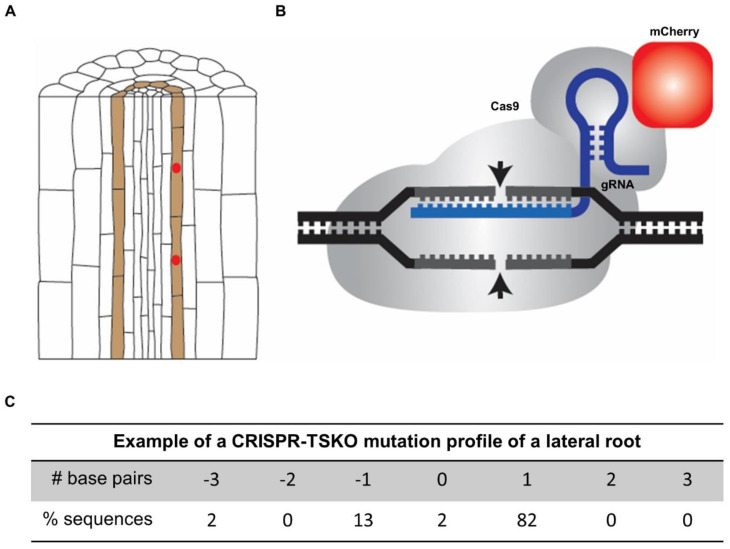
Overview of the lateral root-specific gene knockout system in *Arabidopsis* using p*GATA23*-CRISPR-TSKO. (**A**) Schematic presentation of the specific expression of Cas9 translationally fused to the fluorescent reporter mCherry in lateral root founder cells by using the *GATA23* promoter sequence (indicated in red). (**B**) Schematic presentation of the guide RNA (gRNA)–Cas9 complex. Cas9 is translationally fused to the fluorescent reporter mCherry (indicated in red). (**C**) Example of the mutation profile of a sampled lateral root of a seedling with the p*GATA23*-CRISPR-TSKO construct. The frequency of different alleles of the targeted gene was determined by sanger sequencing and subsequent TIDE analysis [47]. One-base-pair insertions are the predominant outcome.

## Data Availability

Not applicable.

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
