# Peer review of "Lateral Root Initiation and the Analysis of Gene Function Using Genome Editing with CRISPR in Arabidopsis"

_genes, 2021, doi:10.3390/genes12060884_

Round 1

Reviewer 1 Report

Root branching through later root formation is very important for the adaptability of the root system to its environment. Lateral root development is is tightly regulated and auxin serves as a major regulator of lateral root development. Here the authors provided an overview of the visible events, cell cycle regulators and auxin signaling cascades related to the development of lateral root. And CRISPR genome editing system can be used as an important tool to analyse gene function specifically in lateral root initiation and contributed to a better understanding of fundamental developmental processes. The paper is well written and the authors proved a lot of references to supports their viewpoints. I this this paper is suitable for publication in Genes.

Author Response

We thank the reviewer for his/her positive evaluation of our manuscript. No changes were requested by this reviewer

Reviewer 2 Report

The manuscript entitled " Lateral root initiation and the analysis of gene function using genome editing with CRISPR in Arabidopsis " sets out to review lateral root initiation and development and the use of CRISPR technology to study gene function for more understanding of the mechanism. The review is on a topic of relevance and general interest to the readers of the journal. I found the manuscript to be overall well written and felt confident that the authors highlighted important points related to the topic. However, I have a few concerns that should be addressed before publication.

  • The authors are highly recommended to avoid using a personal pronoun (e.g., We, our, etc.); they can use the third party in the past tense's passive voice.
  • The authors need carefully to read the manuscript to correct typos and grammars to improve the manuscript.
  • Any abbreviation must be associated with the full name at the first mention in the abstract and main text, then just use the abbreviation.

Author Response

  • The authors are highly recommended to avoid using a personal pronoun (e.g., We, our, etc.); they can use the third party in the past tense's passive voice. Where possible we avoided using personal pronouns as indicated by the track changes.
  • The authors need carefully to read the manuscript to correct typos and grammars to improve the manuscript. The manuscript has been checked for typos and grammar.
  • Any abbreviation must be associated with the full name at the first mention in the abstract and main text, then just use the abbreviation. Abbreviations were re-checked and full names were used at first mentioning (see track changes).